# Health system responsiveness and its associated factors for delivery care in public health facilities of West Arsi Zone, Oromia, Ethiopia

Negeso Gebeyehu Gejo[1]*, Daniel Yohannes Bedecha[1], Solomon Seyife Alemu[1], Maedot Ariaya Haymete[2], Milion Reta Regasa[1], Abraham Endale Geleta[3], Qorinah Estiningtyas Sakilah Adnani[4]

1 Department of Midwifery, College of Health Sciences, Madda Walabu University, Shashemene Campus, Shashemene, Ethiopia, 2 Virginia Tech Carilion School of Medicine, Roanoke, VA, USA, 3 Department of Public Health, College of Health Sciences, Madda Walabu University, Shashemene Campus, Shashemene, Ethiopia, 4 Department of Public Health, Faculty of Medicine, Universitas Padjadjaran, Bandung, Indonesia

* negiyeman@gmail.com

## Abstract

### Background

Health System Responsiveness is defined as how well the health system meets the legitimate expectations of the population for the non-health enhancing aspects of the health system. As Ethiopia approaches the conclusion of the Health Sector Transformation Plan-II (HSTP-II), generating evidence on health system responsiveness is critical for evaluating progress and guiding future strategies. However, there remains a scarcity of empirical evidence on health system responsiveness, particularly in the context of delivery care services.

### Objective

The primary aim of this study was to assess the health system responsiveness and its associated factors for delivery care in public health facilities of West Arsi Zone, Oromia, Ethiopia.

### Methods

A health facility-based cross-sectional study was conducted among 617 mothers who gave birth in the selected public health facilities of West Arsi Zone, Oromia Region. Data were collected from 24/02/2025–26/04/2025. Systematic random sampling technique was used to approach study participants. Health system responsiveness was measured using eight domains namely dignity, autonomy, confidentiality, communication, prompt attention, social support, choice and basic amenities each item rated on 1–5 scale. Mothers with median score ≥112 were categorized as having good responsiveness performance for delivery care whereas <112 were considered as

**Data availability statement:** All relevant data are within the manuscript and its Supporting Information files.

**Funding:** Madda Walabu University funded the research and it is open for the researchers to publish the manuscript. The funders had no role in study design, data collection and analysis, decision to publish, or preparation of the manuscript.

**Competing interests:** The authors have declared that no competing interests exist.

**Abbreviations:** ANC, Antenatal Care; AOR, Adjusted odds ratio; COR, Crude odds ratio; HSR, Health system responsiveness; HSTP-II, Health Sector Transformation Plan-II; IQR, Inter quartile Range; NGO, Non-governmental Organization; SDGs, Sustainable Development Goals; SPSS, Statistical Package for Social Sciences; UN, United Nations; WHO, World Health Organization.

poor responsiveness for delivery care. Data was entered and analyzed using SPSS version 25. Both bi-variable and multivariable logistic regression analysis were done to identify association between dependent and independent variables. Crude and adjusted odds ratios with respective 95% confidence intervals were computed and statistical significance was declared at p-value <0.05.

## Result

The overall level of good health system responsiveness for delivery care was found to be 51.4% (95% CI 47.4–55.4). The highest and least performance score was reported in the social support domain (62.9%) and choice domain (51.7%) respectively. Adverse neonatal outcome (AOR = 0.50, 95% CI (0.31, 0.82), obstetrics complications (AOR = 0.47, 95% CI (0.26, 0.85), history of admission during current pregnancy (AOR = 0.58, 95% CI (0.35, 0.96) were factors significantly associated with health system responsiveness.

## Conclusions

More than half of the respondents reported that the overall level of health system responsiveness during delivery was good. Adverse neonatal outcome, obstetrics complications and history of admission during current pregnancy had shown statistically significant association with health system responsiveness for delivery care. Enhancing health system responsiveness during delivery needs targeted investments in infrastructure, continuous training for healthcare providers, and tailored support for women who experience adverse neonatal outcomes or obstetric complications. Strengthening these areas is essential for ensuring respectful, timely, and women centered maternity care.

## Introduction

Given the primary purpose of a health system is justice, it is expected to enhance, attain, and distribute health equitably among the entire population in all segments. For this reason, the concept of health system responsiveness was developed and articulated [1]. In addition to equity in the health system's funding, the World Health Organization's (WHO) approach for evaluating the effectiveness of health systems includes health and responsiveness as major outcomes; the extent to which the health system satisfies the public's justifiable expectations regarding its non-health-improving components is known as responsiveness [2,3].

Women hospitalized to obstetrics and gynecology wards have non-medical requirements that should be attended to as quickly as feasible due to their significance. These women have a comfortable and joyful hospital stay when their reasonable expectations are met [4].

The global maternal mortality rate has reduced by 37% since 2000. Despite this, 95,000 women died in pregnancy and delivery in 2017 [5]. Apart from maternal

mortality, each year in poor countries, more than 30 million women do not give birth in a health facility; more than 45 million receive minimal or no antenatal care and more than 200 million women choose to avoid pregnancy but do not utilize contemporary contraceptives [6].

Ethiopia has increased its investments in health facilities and free maternal health services. Skilled delivery grew from 6% in 2005 to 50% in 2019, suggesting that more than half of all deliveries are still done at home. However, there is a disparity in service utilization between urban (72%) and rural (42%) areas [7].

The country has set important health targets in the Health Sector Transformation Plan II (HSTP-II) to bring substantial improvement in maternal and neonatal health. These targets aim to reduce maternal mortality ratio to 279 per 100,000 live births and neonatal mortality to approximately 21 per 1,000 live births by strengthening access to high-quality reproductive, maternal, newborn, and child health services. Besides, as part of this plan, improving non-medical aspects of healthcare such as preferences, desires and moral value were targeted. In addition, updated information regarding responsiveness and its achievement is a focus area of the health sector transformation plan [8,9].

Nonetheless, in Ethiopia, the gap between access to clinical and non-clinical care in the richest and poorest households, particularly skilled attendance during childbirth, is eleven fold higher for the richest than for the poorest. In the same way, urban women have seven times the access to skilled delivery attendants as rural women [8].

Moreover, there is a dearth of evidences on health system responsiveness for delivery care in Ethiopia. Besides, it will be unable to obtain a comprehensive understanding of health responsiveness, particularly at health centers, due to the fact that previous research has focused on hospitals. Nevertheless, this investigation has incorporated health centers to obtain a thorough evaluation of the level of interest. Furthermore, as few months are left for Health Sector Transformation Plan-II to be concluded, evidences form this work are very insightful in informing efforts done thus far. Therefore, the main purpose of this study was to assess health system responsiveness and its associated factors for delivery care in public health facilities of West Arsi Zone, Oromia, Ethiopia.

## Methods

A cross-sectional study was conducted in the selected public health facilities of West Arsi Zone, Oromia, Ethiopia. The Zone is found in Oromia region, southern Ethiopia with the zonal capital city; Shashemene and located 250kms away from Addis Ababa. The zone has 8 public hospitals namely Shashemene Compressive Specialized Hospital, Melka Oda General Hospital, Negelle Arsi Primary Hospital, Dodola General Hospital, Gambo General Hospital, Loke Primary Hospital, Nensebo primary hospital & Kokosa Primary Hospital, and 90 health centers.

The source population consisted of all women who gave birth at public health facilities in West Arsi Zone during the study period. The study population was a subset of this group, comprising randomly selected women who delivered at the selected public health facilities in West Arsi Zone during the study period. All mothers who gave birth in the selected hospitals during the study period were eligible for inclusion in the study. However, women who were critically ill at the time of data collection were excluded.

The sample size was calculated using the double population proportion formula in Open Epi version 2.3.1 statistical software, with the following assumptions: a 95% confidence level, a 1:1 ratio of unexposed to exposed groups, an outcome proportion of 47.9% among the unexposed, 35.93% among the exposed, (taken from a research conducted at public health facilities of Dessie City Administration, South Wollo zone, Ethiopia [10], an adjusted odds ratio (AOR) of 0.61. Finally, after adjusting 10% for non-response, the total sample size was estimated to be 623.

Out of the 8 public hospitals in the Zone, 5 hospitals were selected through simple random sampling technique; these were Shashemene Compressive Specialized Hospital, Melka Oda General Hospital, Dodola General Hospital, Negelle Arsi Primary Hospital and Loke Primary Hospital. Besides, out of 90 health centers found in the Zone, 25 health centers were selected. Because the data collection period covered two months, the average number of deliveries in the preceding two months at each selected facility was used to estimate the caseload. The total sample size was then proportionally

allocated to each facility based on this two-month average caseload, ensuring that facilities with higher delivery volumes contributed a larger number of participants.

Systematic random sampling was used to recruit study participants at each facility. The sampling interval (K) was calculated by dividing the total number of deliveries in the previous year at each respective facility by the sample size allocated to that facility. The first participant was selected by a lottery method from mothers who delivered on the first day of data collection. Thereafter, study participants were selected every second eligible mother was included until the allocated sample size for each facility was reached.

Data were collected using interviewer-administered questionnaire during exit from 24/02/2025–26/04/2025, which is adopted from the health systems responsiveness questionnaire from WHO multi-country studies [1,3]. Besides, secondary data were retrieved from medical records of mothers from 24/02/2025–26/04/2025 to collect information on obstetrics characteristics. The questionnaire consisted three parts which was used to assess socio-demographic characteristics, obstetrics characteristics and eight components (domains) of responsiveness. The validity and reliability of the instrument was assured using pears correlation and Cronbach's alpha co-efficient test respectively.

Data quality was ensured through the use of a well-designed data collection tool. In addition, one day of training was provided to both data collectors and the supervisor. On each day of data collection, the investigators reviewed the completed questionnaires for completeness and consistency. Prior to analysis, the data were edited and cleaned. Furthermore, a pre-test was conducted on 5% of the total sample size (32 women) at Batu General Hospital.

## Measurement

**Health system responsiveness in delivery care**: Eight domains (dignity (5 items), autonomy (3 items), confidentiality (3 items), communication (5 items), prompt attention (3 items), social support (3 items), choice and continuity (3 items) and basic amenities (5 items)) were used to operationalize it. These eight domains of health system responsiveness are important in delivery care, as they collectively ensure that women receive respectful, timely, and person-centered services that directly influence their safety, satisfaction, and overall birth experience.

The score per domain was given in 1–5 scale scores (1- Strongly disagree, 2- Disagree, 3- Neutral, 4- Agree, and 5- Strongly agree). For each domain, the five options answers were grouped into binary categories; 'good responsiveness' and 'poor responsiveness'. Then, the total median score for overall Health system responsiveness (HSR) was calculated from the median of the respective domains because the data was not normally distributed. The median score for the overall HSR was 112. Mothers who had score above the median (≥112) were considered as having good responsiveness performance for delivery care whereas an overall score <112 median score were considered as poor responsiveness for delivery care.

The outcome variable was dichotomized using the median because the distribution of responsiveness scores was not normal. Median based dichotomization is therefore an appropriate and statistically robust method for skewed data, allowing consistent classification of respondents into higher versus lower responsiveness categories while minimizing the effect of outliers.

The WHO Health System Responsiveness Questionnaire is a globally validated instrument developed through extensive multi-country field testing across diverse cultural and health system contexts. Because of its strong conceptual validity, cross-cultural applicability, and standardized item structure, the instrument is recommended for use without modification to maintain comparability with global evidence and to preserve the validity of the tool.

The Cronbach's alpha values for dignity, autonomy, confidentiality, communication, prompt attention, social support, choice, and quality of basic amenities were 0.789, 0.890, 0.868, 0.813, 0.807, 0.781, 0.876, and 0.788, respectively.

## Data analysis

After recording, the collected data was entered and analyzed using SPSS version 25.0 software.

Descriptive statistics such as frequencies, proportions, inter-quartile range and median were calculated to describe socio-demographic and obstetrics characteristics of the study population was displayed using tables and graphs.

Both bi-variable and multivariable logistic regression analysis was used to determine the association of each independent variable with the dependent variable. Initially, a variable with p < 0.30 at bi-variable logistic regression model was taken in to multivariable logistic regression model. A p-value threshold of less than 0.30 was used because, in most of epidemiological and public health research it is common approach to take a relatively higher p-value threshold to avoid excluding potentially important variables that may become significant after adjustment for confounders. Both crude and adjusted odds ratio with respective 95% confidence intervals and p value <0.05 was used to measure the strength of association between dependent and independent variables.

The dependent variable (HSR) was computed by adding all the eight domains separately first and calculating the median. The medians of all the respective domains were added and finally the median value was taken to determine HSR. The median score for the overall HSR was 112. Mothers who had score above the median (≥112) was taken as having good HSR and below the median (<112) were taken as having poor HSR. Mothers who reported good HSR were coded "1" and mothers who reported poor HSR were coded "0".

Multi-collinearity between the independent variables was assessed using variance inflation factor (VIF). The VIF values for husband's educational level, place of residence, parity, time of delivery, duration of labour, length of hospital stay, adverse neonatal outcome, obstetric complications, and admission history during the current pregnancy were 1.103, 1.074, 1.037, 1.034, 1.086, 1.191, 1.536, 1.401, and 1.388, respectively. Thus, all the independent variables had VIF value less than 5, suggesting there was no multi-collinearity problem.

### Ethics approval and consent to participate

Ethical approval was taken from Madda Walabu University, Shashemene campus research review and ethics committee before commencing data collection with ref number RCSTT/71/2055. Support letters were written to respective health institutions.

### Written informed consent

Written informed consent was obtained from all study participants prior to their enrollment in the study. For participants who were unable to write, a right thumbprint was taken as a signature. Participants were informed about the purpose, procedures, potential risks, and benefits of the study, as well as their right to withdraw at any time without any consequences. The information sheet was provided in a language they understood, and adequate time was given for questions and clarification. Only those who voluntarily signed the consent form were included in the study. For participants under 18 years or those unable to provide consent independently, consent was obtained from a parent or legal guardian. The information provided by each respondent was kept confidential and de-identified and de-linked and kept in a secure location.

### Inclusivity in global research

Additional information regarding the ethical, cultural, and scientific considerations specific to inclusivity in global research is included.

## Results

### Socio-demographic characteristics of the study participants

A total of 617 women were participated in the current study, making a response rate of 99.03%. The median age of the respondents was 28 (IQR 25, 30). The age ranges from 17 to 45 years. All participants were married. Nearly one-third of study subjects 202 (32.7%) had attended primary education whereas nearly one-third study subjects' husbands 190

(30.8%) had attended secondary education. Nearly three-fourth of participants 452 (73.3%) were housewives. More than half of study participants 408 (66.1%) were urban residents (Table 1).

## Obstetrics characteristics of the study participants

Slightly, more than one-tenth of women 81 (13.1%) were primiparous. 490 (91.4%) women had history of institutional delivery. Nearly one-third of study of the study subjects 166 (31.0%) had history of adverse maternal & perinatal outcome. More than three-fourth of mothers 534 (86.5%) had antenatal care (ANC) follow-up. Nearly one-third of participants 199 (32.3%) reported that their current pregnancy was not planned. Concerning perinatal care path way, nearly half of study subjects 294 (47.6%) their entire care was at a single health facility (Table 2).

More than three-fourth of women 545 (88.3%) had given birth vaginally. More than three-fourths of the births were attended by midwives, 501 (81.2%) (Fig 1). In more than half of study subjects 367 (59.5%) non-emergency intervention were done during labour. The duration of labour was ≥ 12 hours in nearly three-fourth of mothers 430 (69.7%). Nearly one-third of study participants 179 (29.0%) had stayed for more than 24 hours in the hospital. Nearly, one-quarter of mothers 141 (22.9%) had faced adverse neonatal outcome in the current pregnancy while slightly more than one-third of mothers 86 (13.9%) had developed obstetrics complications (Table 3).

**Table 1. Socio-demographic characteristics of mothers who gave birth in public health facilities of West Arsi Zone, Oromia, Ethiopia, 2025 (n = 617).**

| Variable | Category | Frequency | Percentage (%) |
|---|---|---|---|
| Maternal age | 15-19 | 28 | 4.5 |
| | 20-24 | 114 | 18.5 |
| | 25-29 | 248 | 40.2 |
| | 30-34 | 200 | 32.4 |
| | >=35 | 27 | 4.4 |
| Mother's educational level | Can't read & write | 76 | 12.3 |
| | Read & write only | 94 | 15.2 |
| | Primary education | 202 | 32.7 |
| | Secondary education | 177 | 28.7 |
| | Above Secondary | 68 | 11.0 |
| Husband's educational level | Can't read & write | 40 | 6.5 |
| | Read & write only | 172 | 27.9 |
| | Primary education | 81 | 13.1 |
| | Secondary education | 190 | 30.8 |
| | Above Secondary | 134 | 21.7 |
| Mother's occupation | Government employee | 49 | 7.9 |
| | Housewife | 452 | 73.3 |
| | Merchant | 84 | 13.6 |
| | NGO | 8 | 1.3 |
| | Student | 11 | 1.8 |
| | Farmer | 11 | 1.8 |
| | Daily laborer | 2 | 0.3 |
| Place of residence | Rural | 209 | 33.9 |
| | Urban | 408 | 66.1 |

**Table 2. Obstetrics characteristics of mothers who gave birth in public health facilities of West Arsi Zone, Oromia, Ethiopia, 2025.**

| Variable | Category | Frequency | Percentage (%) |
|---|---|---|---|
| Parity (n = 617) | 1 | 81 | 13.1 |
| | 2-4 | 101 | 16.4 |
| | ≥5 | 435 | 70.5 |
| History of Institutional Delivery (n = 536) | Yes | 490 | 91.4 |
| | No | 46 | 8.6 |
| History of adverse maternal & perinatal outcome (n = 536) | Yes | 166 | 31.0 |
| | No | 370 | 69.0 |
| ANC follow-up(n = 617) | Yes | 534 | 86.5 |
| | No | 83 | 13.5 |
| No of ANC Contacts (n = 534) | 1-2 | 85 | 13.8 |
| | 3-4 | 2 | 0.3 |
| | ≥5 | 530 | 85.9 |
| Planned Pregnancy (n = 617) | Yes | 418 | 67.7 |
| | No | 199 | 32.3 |
| Perinatal health care path (n = 617) | No ANC, referred during labour | 49 | 7.9 |
| | Referred during pregnancy | 168 | 27.2 |
| | One health facility entirely, not referred | 294 | 47.6 |
| | Referred during parturition | 106 | 17.2 |

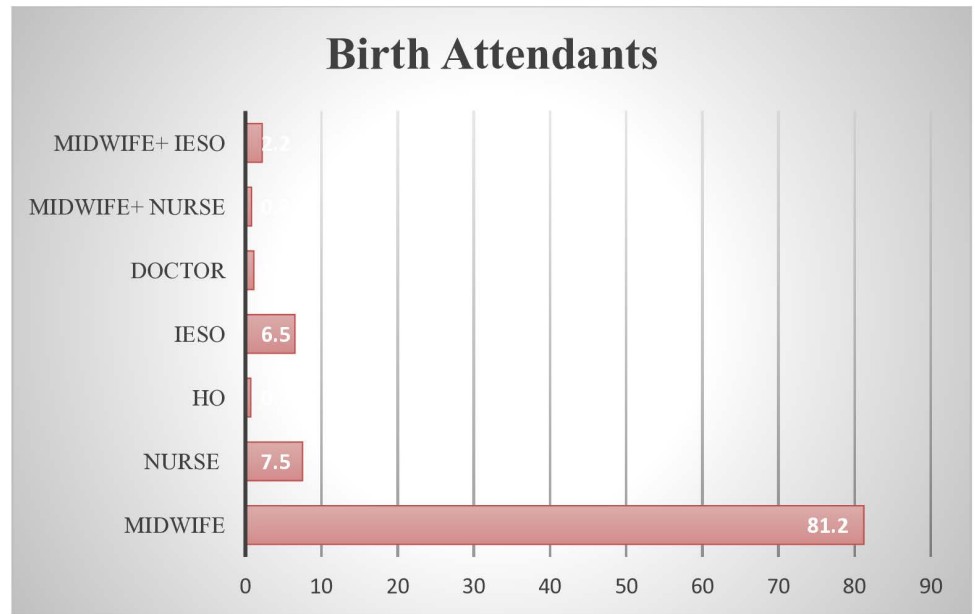

**Fig 1. Birth attendants by profession mothers who gave birth in public health facilities of West Arsi Zone, Oromia, Ethiopia, 2025.**

**Table 3. Labour and delivery related characteristics of mothers who gave birth at public health facilities in West Arsi Zone (n=617).**

| Variable | Category | Frequency | Percentage (%) |
|---|---|---|---|
| Mode of delivery (n=617) | Spontaneous vaginal delivery | 496 | 80.4 |
| | Cesarean section | 64 | 10.4 |
| | Instrumental | 57 | 9.2 |
| Time of Delivery (n=617) | Day | 442 | 71.6 |
| | Night | 175 | 28.4 |
| Day of Delivery (n=617) | Monday to Friday | 505 | 81.8 |
| | Weekend | 112 | 18.2 |
| Duration of labour (n=617) | <12 hours | 187 | 30.3 |
| | ≥12 hours | 430 | 69.7 |
| Length of hospital stay (n=617) | <24 hours | 438 | 71.0 |
| Adverse neonatal outcome(n=617) | ≥24 hours | 179 | 29.0 |
| | No | 476 | 77.1 |
| Obstetrics complication (n=617) | Yes | 141 | 22.9 |
| | No | 531 | 86.1 |
| Onset of labour (n=617) | Elective Cesarean section (prior to labour) | 47 | 7.6 |
| | Induced | 25 | 4.1 |
| | Spontaneous | 545 | 88.3 |

## Health service accessibility related variables

More than three-fourths of the women, 489 (79.3%), traveled less than one hour to reach a health facility. Almost all mothers, 610 (98.9%), reported that transportation was available to reach the health facilities.

## Health system responsiveness

Health system responsiveness for delivery care was measured by eight domains including dignity, autonomy, confidentiality, communication, prompt attention, social support, choice and qualities of amenities using likert scale. First, the median value of each domain was determined because the data for all domains is not normally distributed. Lastly, since the data was not normally distributed, the overall health system responsiveness for delivery care was computed using the median value.

Thus, the overall level of good health system responsiveness for delivery care was found to be 51.4% (95% CI 47.4–55.4). The highest and least performance score of good health system responsiveness was reported in the social support domain (62.9%) and choice domain (51.7%) respectively (Fig 2) (Table 4).

## Factors associated with health system responsiveness

Husband's educational level, place of residence, parity, time of delivery, duration of labour length of hospital stay, adverse neonatal outcome, maternal complication and admission history during current pregnancy were factors which had p-values less than 0.30 during bi-variable logistic regression analysis. After controlling for possible confounders at multi-variable logistic regression model, adverse neonatal outcome, maternal complication and admission history during current pregnancy were statistically significantly associated with perinatal health outcomes at p-value <0.05. Hosmer and Lemeshow goodness of fit statistics was done to check for model fitness and it was found to be a good fit (Chi-square=9.57 & P=0.214)

The odds of reporting good health system responsiveness for delivery care among women who had adverse neonatal outcome were 50% times less likely than those without adverse neonatal outcome(AOR=0.50, 95% CI (0.31, 0.82). Similarly,

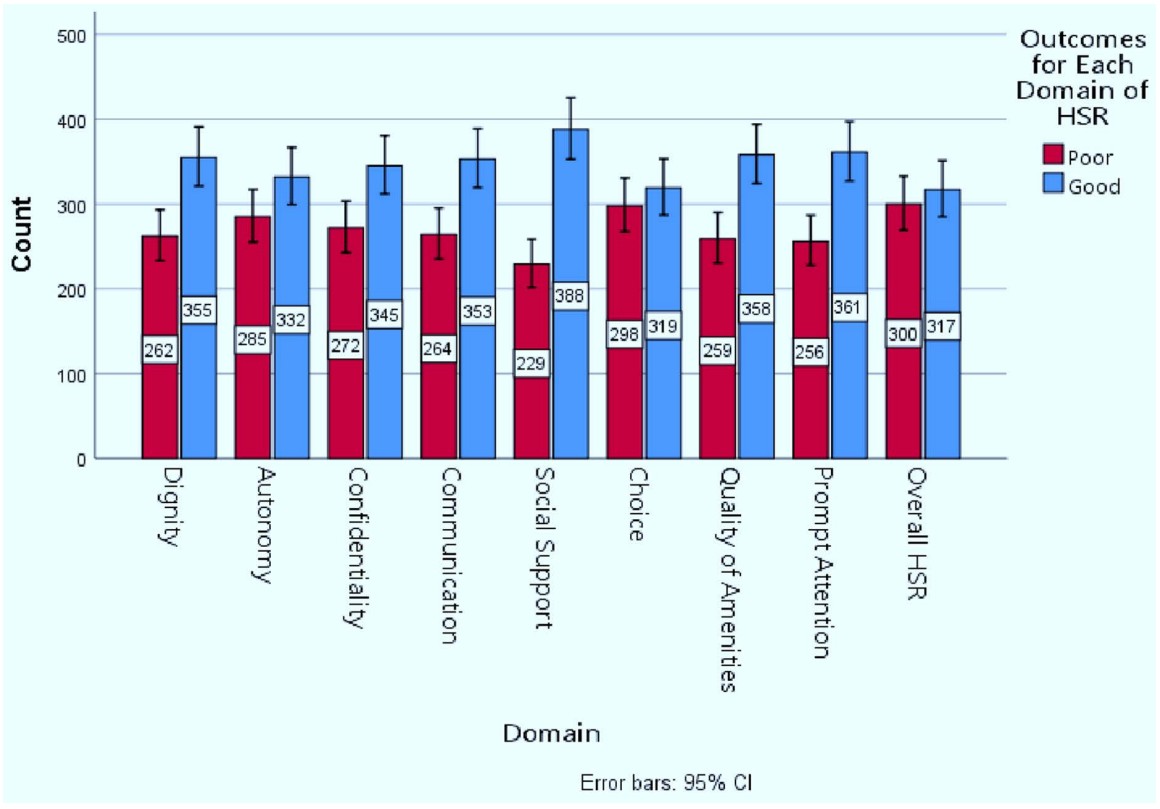

**Fig 2. Score for the respective domains of health system responsiveness with overall score among women who gave birth at public health facilities of West Arsi Zone, Oromia Region, Ethiopia, 2025.**

**Table 4. Health system responsiveness for the respective domains among mothers who gave birth in public health facilities of West Arsi Zone, Oromia, Ethiopia, 2025 (n=617).**

| Variable | Category | Frequency | Percentage |
|---|---|---|---|
| Dignity | Poor | 262 | 42.5 |
| | Good | 355 | 57.5 |
| Autonomy | Poor | 285 | 46.2 |
| | Good | 332 | 53.8 |
| Confidentiality | Poor | 272 | 44.1 |
| | Good | 345 | 55.9 |
| Communication | Poor | 264 | 42.8 |
| | Good | 353 | 57.2 |
| Prompt Attention | Poor | 256 | 41.5 |
| | Good | 361 | 58.5 |
| Social Support | Poor | 229 | 37.1 |
| | Good | 388 | 62.9 |
| Quality of Basic Amenities | Poor | 259 | 42.0 |
| | Good | 358 | 58.0 |
| Overall Health System Responsiveness | Poor | 300 | 48.6 |
| | Good | 317 | 51.4 |

the odds of reporting good health system responsiveness among women who had obstetrics complications were 53% times less likely than those without obstetrics complications (AOR = 0.47, 95% CI (0.26, 0.85). Likewise, the odds of reporting good health system responsiveness among women who had history of admission during current pregnancy were 42% times less likely than those who were not admitted during current pregnancy(AOR = 0.58, 95% CI (0.35, 0.96) (Table 5).

## Discussion

The present study assessed health system responsiveness for delivery care at the public health facilities of West Arsi Zone. According to the current study, more than half of mothers 317 (51.4%) reported the overall performance of health system responsiveness for delivery care as good. However, the finding highlighted that significant proportion of mothers reported poor health system responsiveness for delivery care. Besides, dignity was upheld for 57.5%, autonomy for 53.8%, confidentiality for 55.9%, communication for 57.2%, prompt attention for 58.5%, social support for 62.9%, choice for 51.7% and quality of basic amenties for 58%.

The finding of health system responsiveness magnitude in the present study is comparable with the study conducted in Hadiya zone, in which 53.0% of users reported good responsiveness for delivery care. However, the responsiveness performance in maternity care was high for the dignity domain 77% which is inconsistent with the present study [11].

**Table 5. Bi-variable & multivariable logistic regression analysis for the factors associated with HSR for delivery care at public health facilities of West Arsi Zone, Oromia, Ethiopia.**

| Variable | Category | HSR | | COR (95% CI) | AOR (95% CI) | P-value |
|---|---|---|---|---|---|---|
| | | Good No. (%) | Poor No. (%) | | | |
| Husband's educational level | Can't read & write | 20 (50) | 20 (50) | 1 | 1 | 1 |
| | Read & write only | 48 (59.3) | 33 (40.7) | 1.46 (0.68, 3.12) | 1.35 (0.61, 2.99) | 0.463 |
| | Primary education | 69 (40.1) | 103 (59.1) | 0.67 (0.34, 1.34) | 0.56 (0.27, 1.15) | 0.116 |
| | Secondary education | 110 (57.9) | 80 (42.1) | 1.38 (0.69, 2.72) | 1.21 (0.59, 2.48) | 0.592 |
| | Above Secondary | 70 (52.2) | 64 (47.8) | 1.09 (0.54, 2.22) | 1.12 (0.53, 2.37) | 0.755 |
| Place of residence | Rural | 110 (52.6) | 99 (47.4) | 1 | 1 | 1 |
| | Urban | 207 (50.3) | 201 (49.3) | 0.93 (0.66, 1.23) | 0.89 (0.61, 1.31) | 0.574 |
| Parity | 1 | 37 (45.7) | 44 (54.3) | 1 | 1 | 1 |
| | 2-4 | 51 (50.5) | 50 (49.5) | 1.21 (0.68, 2.18) | 1.34 (0.72, 2.49) | 0.352 |
| | ≥5 | 229 (52.6) | 206 (47.4) | 1.32 (0.82, 2.13) | 1.43 (0.86, 2.38) | 0.166 |
| Time of Delivery | Day | 231 (52.3) | 211(47.7) | 1 | 1 | 1 |
| | Night | 86 (49.1) | 89 (50.1) | 0.88 (0.62, 1.25) | 0.92 (0.63, 1.35) | 0.679 |
| Duration of labour | <12 hrs. | 239 (54.6) | 199 (45.4) | 1 | 1 | 1 |
| | ≥12 hrs. | 78 (43.6) | 101(56.4) | 0.78 (0.55, 1.11) | 0.74 (0.51, 1.07) | |
| Length of hospital stay | <24 hrs. | 104 (55.6) | 83 (44.4) | 1 | 1 | 1 |
| | ≥24 hrs. | 213 (49.5) | 217 (50.5) | 0.64 (0.45, 0.91) | 0.85 (0.56, 1.27) | 0.421 |
| Adverse neonatal outcome | No | 275 (57.8) | 201 (42.2) | 1 | 1 | 1 |
| | Yes | 42 (29.8) | 99 (70.2) | 0.31 (0.21, 0.47) | **0.50 (0.31, 0.82)*** | **0.006*** |
| Obstetrics complications | No | 294 (55.4) | 237 (44.6) | 1 | 1 | 1 |
| | Yes | 23 (26.7) | 63 (73.3) | 0.29 (0.18, 0.49) | **0.47 (0.26, 0.85)*** | **0.012*** |
| Admission history during current pregnancy | No | 280 (56.1) | 219 (43.9) | 1 | 1 | 1 |
| | Yes | 37 (31.4) | 81 (68.6) | 0.36 (0.23, 0.55) | **0.58 (0.35, 0.96)*** | **0.034*** |

(*shows statistical significance at p-value <0.05, 1_shows reference category).

However, it is slightly higher than the study carried out in Dessie City Administration, South Wollo zone, Ethiopia (45.8%). The highest (74.2%) and lowest (45.8%) rated domains were dignity and basic amenity, respectively [10]. Likewise, it is slightly higher than the study conducted in Amhara region in areas affected by conflict (45.11%) [12].

The variation could be attributed to the study setting and period. For instance, the Dessie City study [10] was carried in 2022. This suggests that as time passes, new directions and strategies will be implemented by the government improving the quality of health care delivery system, and the degree of compassionate, respectful maternity care may improve. On the other hand, the study from the Amhara region [12] was conducted in conflict zones where people were displaced, which may have had an impact on their mental health and level of satisfaction, and health facilities may have had limited human and medical resources, reducing the responsiveness of the health system.

The finding of the current study is lower than a study conducted in Asagirt District, North Shewa Zone, Ethiopia (66.2%).Confidentiality and dignity were the highest responsive domains [13]. Similarly, the finding of present study is lower than a cross-sectional survey conducted in Spain (77.4%) [14], Bangladesh (67%) [15], Iran (58.4%) [16], Tanzania (68%) [17].

The variation could be due to difference socio-economic factors, study setting and study population. For example, the study in Bangladesh [15] addressed private settings, whereas the current study only examined public health facilities. Similarly, the study in Spain [14] involved people who had mental illnesses whereas the present study included mothers who gave birth at the respective health facilities. Similarly, the study in Iran included patients from all wards, with the exception of emergency and neurology units.

The present study revealed that choice (51.7%) and autonomy (53.8%) to be lowest health system responsiveness domains while social support scored the highest responsiveness (62.9%). Choice and autonomy domains are low, despite policy emphasis on patient centered care within Ethiopia's Health Sector Transformation Plan (HSTP) [8]. This finding suggests a persistent gap between policy intentions and actual service delivery practices. Although national guidelines advocate for respectful care, shared decision making, and the right of patients to choose providers and participate in treatment decisions, implementation at facility level remains limited. This is in line with study conducted in Amhara region [12] where autonomy, choice, and immediate attention domains, responsiveness performed lowest, at 35.5%, 49.4%, and 52.0%, respectively. Similarly, the finding of the present study is consistent with the study conducted in Iran where social support networks dimension had the highest responsiveness, with a mean score of $3.95 \pm 0.92$ whereas the autonomy and choice of therapist dimensions had the lowest responsiveness, with mean scores of $2.81 \pm 0.71$ and $2.70 \pm 0.33$, respectively [18].

In the present study, socio-demographic factors have not shown statistically significant association with health system responsiveness. One possible explanation for the non-significant associations in our study is sample homogeneity. Most of the study participants share similar socio-demographic characteristics such as comparable educational backgrounds, residency, and access to services. Thus, the variability needed to detect meaningful differences becomes limited. This is supported by studies done in Dessie City [10] and Shewarobit [19]. However, it is in contrary with the study done in Hadiya Zone [11], Amhara region [12] and Southwest of Ethiopia [20], Mashhad, North-East Iran [21], Spain [14] and South Africa [22].

The odds of reporting good health system responsiveness for delivery care among women who had adverse neonatal outcome were 50% times less likely than those without adverse neonatal outcome. This could be due to detrimental impacts on their neonate, or women may be dissatisfied with the health care delivery system in general, negatively affecting health system responsiveness.

Furthermore, if the detrimental effects on occurred, the health-care system's responsiveness will be hampered further. This means that enough attention, sympathy, and reassurance must be given to these mothers, to enhance health-care system responsiveness. This finding is in line with study done in Dessie City [10], Hadiya Zone [11], and Netherlands [23].

Similarly, the odds of reporting good health system responsiveness among women who had obstetrics complications were 53% times less likely than those without maternal complication. This could be due to disappointment with the

complications that occurred; mothers may lose interest in the health care delivery system, reducing the health system's responsiveness. Furthermore, if the complications are caused by a shortage of medical resources, inexperience of the health care practitioner, or mothers' poor compliance, it will have a negative impact on the health system responsiveness. This finding is consistent with study done in Dessie City [10], Hadiya Zone [11], and Netherlands [23].

Likewise, the odds of reporting good health system responsiveness among women who had history of admission during current pregnancy were 42% times less likely than those who were not admitted during current pregnancy. This could be because women will be admitted to the obstetrics ward as a result of obstetric problems, in order to manage and prevent further complications on the mother and fetus. As a result, unforeseen complications, due to non-compliance towards the medical treatment, medical error, inexperience of health care providers, lack of medical resources including maternal and neonatal complications, may occur. This will have negative impacts the health system responsiveness. However, this finding is in contrary with the study conducted in Amhara region where mothers admitted to a health facility before delivery reported higher levels of health system responsiveness [12].

## Conclusions

In the present study, more than half of the respondents reported the overall level of health system responsiveness during delivery as good. However, a significant proportion of mothers perceived the responsiveness of the health system as poor. Adverse neonatal outcomes, obstetric complications, and a history of admission during the current pregnancy were significantly associated with health system responsiveness. These findings highlight critical areas where maternal care services can be strengthened to meet the expectations and needs of women during delivery.

Based on these findings, it is recommended that zonal health departments and health facilities enhance responsiveness across all domains by improving infrastructure and providing targeted training for healthcare providers. The low scores observed in the autonomy and choice domains highlight critical gaps in women centered care, indicating the need for targeted interventions. First, capacity building initiatives must be strengthened by integrating a wide range of training on informed decision making, women communication, and respectful care. Such training will equip health care providers with the skills required to involve women in their own health care, respect their choices, and provide clear, comprehensible health care information.

Second, health facility infrastructure and service organization require improvement to enhance women's exercise choice. This includes expanding the availability of essential services, increasing number of qualified staff, and ensuring continuity of care. Strengthening referral systems, improving appointment scheduling system, and ensuring privacy during examinations and procedures can further reinforce women's sense of control and involvement in their care.

Moreover, special attention has to be given to women who experience adverse neonatal outcomes, obstetric complications, or hospital admissions during pregnancy, as these factors negatively impact perceived responsiveness. Integrating responsiveness indicators into routine maternal health quality assessments has to be considered. Additionally, future research should explore health system responsiveness from the perspective of healthcare providers to provide a more comprehensive understanding and guide policy and practice improvements.

This study has several methodological limitations. Firstly, interviewer administered data collection may have introduced social desirability bias, whereby study participants made positive responses to appear acquiescent or avoid offending the interviewer. This may have inflated responsiveness scores. On the other hand, women were included from health facilities only, inherently excluding women who gave birth at home. Given that home deliveries are often associated with lower satisfaction levels, different expectations, or poorer experiences with the health system, their exclusion may limit the generalizability of the results.

Despite these limitations, the findings provide important insights for strengthening women centered care and improving responsiveness within Ethiopia's health system.

## Supporting information

**S1 File. Data set.**
(XLSX)

**S2 File. English Questionnaire.**
(DOCX)

**S3 File. Afan Oromo Questionnaire.**
(DOCX)

## Acknowledgments

We are sincerely grateful to Madda Walabu University for their support, and we extend our appreciation to the data collectors, supervisors, health facility managers, and all study participants, without whom this study would not have been possible."

## Author contributions

**Conceptualization:** Negeso Gebeyehu Gejo, Daniel Yohannes Bedecha.

**Data curation:** Negeso Gebeyehu Gejo, Solomon Seyife Alemu, Abraham Endale Geleta.

**Formal analysis:** Negeso Gebeyehu Gejo, Daniel Yohannes Bedecha, Solomon Seyife Alemu, Maedot Ariaya Haymete.

**Funding acquisition:** Negeso Gebeyehu Gejo.

**Investigation:** Negeso Gebeyehu Gejo, Milion Reta Regasa, Abraham Endale Geleta, Qorinah Estiningtyas Sakilah Adnani.

**Methodology:** Negeso Gebeyehu Gejo, Daniel Yohannes Bedecha, Solomon Seyife Alemu, Milion Reta Regasa, Abraham Endale Geleta.

**Resources:** Negeso Gebeyehu Gejo, Daniel Yohannes Bedecha, Solomon Seyife Alemu, Maedot Ariaya Haymete, Abraham Endale Geleta, Qorinah Estiningtyas Sakilah Adnani.

**Software:** Negeso Gebeyehu Gejo, Solomon Seyife Alemu, Milion Reta Regasa.

**Supervision:** Negeso Gebeyehu Gejo, Milion Reta Regasa.

**Validation:** Negeso Gebeyehu Gejo, Daniel Yohannes Bedecha, Maedot Ariaya Haymete.

**Visualization:** Negeso Gebeyehu Gejo.

**Writing – original draft:** Negeso Gebeyehu Gejo.

**Writing – review & editing:** Qorinah Estiningtyas Sakilah Adnani.

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
