## [Decision Letter · Decision Letter 0]

31 Oct 2025

Dear Dr. Gejo,

Thank you for submitting your manuscript to PLOS ONE. After careful consideration, we feel that it has merit but does not fully meet PLOS ONE’s publication criteria as it currently stands. Therefore, we invite you to submit a revised version of the manuscript that addresses the points raised during the review process.

The topic is important and timely; the study has potential to contribute meaningful Ethiopian-context evidence, especially if it clearly demonstrates domain-specific gaps and actionable implications.However, the manuscript in its current form requires substantial revision to improve clarity, methodological rigor, and reporting quality across sections.Key issues requiring revisions (by section)Abstract Need precise reporting of key findings (e.g., avoid ambiguous phrases like “relatively higher than similar studies”).Rephrase unclear phrasing (e.g., “crucial in impact health system responsiveness”).If possible, list the exact HSR domains assessed.

Introduction Condense global context to foreground Ethiopia-specific gaps.Provide justification for why each of the eight HSR domains matters in delivery care.Clearly state the study’s novelty (e.g., inclusion of health centers; timing near the end of HSTP-II) and situate it within recent literature on HSR and respectful maternity care.

Methods

Sampling: Specify how systematic sampling was implemented; show how K and proportional allocation across facilities were calculated.

Measurement: Justify median-based dichotomization; explain use of the WHO HSR tool without local adaptation; report Cronbach’s alpha for each domain.

Data analysis: Justify the p<0.30 threshold for variable selection; report VIF values.

Ethics: Include the ethics approval number.

Results

Improve table formatting for clarity (percentages and frequencies aligned); ensure all figures have confidence intervals or error bars.

Consider a domain-wise summary table of responsiveness percentages.

Report Hosmer-Lemeshow test results in Results (not only Methods).

Discussion Deepen cross-study comparisons (e.g., why autonomy and choice remain low despite policy efforts).

Avoid speculative statements not supported by data (e.g., “medical error” as a reason).

Explain non-significant socio-demographic factors; discuss potential sample homogeneity.

Provide concrete policy implications and a dedicated section for recommendations (training, infrastructure, governance).

Expand strengths and limitations; acknowledge social desirability bias and facility-based sampling limitations (potential exclusion of home deliveries).

Conclusion Avoid vague phrases; anchor conclusions to the collected data.

Explicitly distinguish empirical findings from recommendations.

Language and presentation

Address grammar and awkward phrasing throughout; perform a thorough language edit.

Major Revision Required with resubmission. The manuscript shows promise but currently lacks the rigor, transparency, and clarity necessary for publication. A revised manuscript that addresses the above methodological, reporting, and language issues, accompanied by a detailed point-by-point response to reviewers, would be eligible for further evaluation.

We look forward to receiving your revised manuscript.

Kind regards,

Engidaw Fentahun Enyew

Academic Editor

PLOS ONE

**Journal Requirements:**

1. When submitting your revision, we need you to address these additional requirements. Please ensure that your manuscript meets PLOS ONE's style requirements, including those for file naming. The PLOS ONE style templates can be found at https://journals.plos.org/plosone/s/file?id=wjVg/PLOSOne_formatting_sample_main_body.pdf and https://journals.plos.org/plosone/s/file?id=ba62/PLOSOne_formatting_sample_title_authors_affiliations.pdf 2. Please include a complete copy of PLOS’ questionnaire on inclusivity in global research in your revised manuscript. Our policy for research in this area aims to improve transparency in the reporting of research performed outside of researchers’ own country or community. The policy applies to researchers who have travelled to a different country to conduct research, research with Indigenous populations or their lands, and research on cultural artefacts. The questionnaire can also be requested at the journal’s discretion for any other submissions, even if these conditions are not met. Please find more information on the policy and a link to download a blank copy of the questionnaire here: https://journals.plos.org/plosone/s/best-practices-in-research-reporting. Please upload a completed version of your questionnaire as Supporting Information when you resubmit your manuscript. 3. Please amend either the abstract on the online submission form (via Edit Submission) or the abstract in the manuscript so that they are identical. 4. We note that this data set consists of interview transcripts. Can you please confirm that all participants gave consent for interview transcript to be published? If they DID provide consent for these transcripts to be published, please also confirm that the transcripts do not contain any potentially identifying information (or let us know if the participants consented to having their personal details published and made publicly available). We consider the following details to be identifying information:- Names, nicknames, and initials- Age more specific than round numbers- GPS coordinates, physical addresses, IP addresses, email addresses- Information in small sample sizes (e.g. 40 students from X class in X year at X university)- Specific dates (e.g. visit dates, interview dates)- ID numbers Or, if the participants DID NOT provide consent for these transcripts to be published:- Provide a de-identified version of the data or excerpts of interview responses- Provide information regarding how these transcripts can be accessed by researchers who meet the criteria for access to confidential data, including:a) the grounds for restrictionb) the name of the ethics committee, Institutional Review Board, or third-party organization that is imposing sharing restrictions on the datac) a non-author, institutional point of contact that is able to field data access queries, in the interest of maintaining long-term data accessibility.d) Any relevant data set names, URLs, DOIs, etc. that an independent researcher would need in order to request your minimal data set. For further information on sharing data that contains sensitive participant information, please see: https://journals.plos.org/plosone/s/data-availability#loc-human-research-participant-data-and-other-sensitive-data If there are ethical, legal, or third-party restrictions upon your dataset, you must provide all of the following details (https://journals.plos.org/plosone/s/data-availability#loc-acceptable-data-access-restrictions):a) A complete description of the datasetb) The nature of the restrictions upon the data (ethical, legal, or owned by a third party) and the reasoning behind themc) The full name of the body imposing the restrictions upon your dataset (ethics committee, institution, data access committee, etc)d) If the data are owned by a third party, confirmation of whether the authors received any special privileges in accessing the data that other researchers would not havee) Direct, non-author contact information (preferably email) for the body imposing the restrictions upon the data, to which data access requests can be sent 5. If the reviewer comments include a recommendation to cite specific previously published works, please review and evaluate these publications to determine whether they are relevant and should be cited. There is no requirement to cite these works unless the editor has indicated otherwise. 

Reviewers' comments:

**Comments to the Author**

1. Is the manuscript technically sound, and do the data support the conclusions?

Reviewer #1: Yes

Reviewer #2: Yes

2. Has the statistical analysis been performed appropriately and rigorously?

Reviewer #1: Yes

Reviewer #2: Yes

3. Have the authors made all data underlying the findings in their manuscript fully available?

Reviewer #1: Yes

Reviewer #2: Yes

4. Is the manuscript presented in an intelligible fashion and written in standard English?

Reviewer #1: Yes

Reviewer #2: Yes

**Reviewer #1: ** The manuscript is suitable for PLOS ONE because:

Public Health Relevance: It examines Health System Responsiveness (HSR), a key non-clinical quality component defined by the WHO, which is critical for achieving universal health coverage (UHC).

Methodology: The quantitative, cross-sectional design utilizing a structured questionnaire is a standard, appropriate approach for assessing HSR.

Gap Filling: The study specifically aims to provide empirical evidence on HSR for delivery care, which the authors correctly note is scarce in the literature for this region.

The manuscript should be Accepted Pending Major Revisions. Attached

**Reviewer #2:**  The manuscript addresses an important topic, health system responsiveness for delivery care in Ethiopia, which is relevant for maternal health policy and practice. The study is timely given the conclusion of HSTP-II and global emphasis on respectful maternity care. However, several sections require clarification, methodological strengthening, and language refinement.

Abstract (pp. 10–11, lines 26–57)

The abstract summarizes the study well but could be more precise in reporting key findings. For example, clarify what “relatively higher than similar studies”

The phrase “crucial in impact health system responsiveness” (line 55) is unclear—please rephrase for clarity.

May include the exact domains assessed (if it will not impact word count) in the abstract for completeness.

Introduction (pp. 11–12, lines 59–102)

The introduction provides extensive global context but could be condensed. Lines 70–81 on SDGs and Agenda 2063 can be summarized to focus more on Ethiopia-specific gaps.

The definition of HSR is clear, but the manuscript could benefit from a brief explanation of why each of the eight domains (e.g., dignity, autonomy, choice) is critical in delivery care. where should this be addressed

Clearly articulate the novelty of this study compared to previous Ethiopian studies (e.g., inclusion of health centers, timing near HSTP-II conclusion).

Include recent literature on health system responsiveness and respectful maternity care.

Methods (pp. 12–15, lines 103–193)

Sampling: Lines 116–130 describe sampling but lack detail on how systematic sampling was implemented. Clarify calculation of K and proportional allocation across facilities.

Measurement: Lines 144–155—justify dichotomization using median and explain why WHO responsiveness tool was used without local adaptation. Provide Cronbach’s alpha values for each domain.

Data Analysis: Lines 162–166—justify p<0.30 threshold for variable selection. Report VIF values explicitly.

Include ethics approval number (line 178).

Results (pp. 16–22, lines 194–303)

Tables 1–3 (pp. 17–20) need improved formatting for clarity—align percentages and frequencies.

Figure 2 (p. 21) should include confidence intervals or error bars for domain scores.

Consider adding a summary table for domain-wise responsiveness percentages (lines 277–279).

Report Hosmer-Lemeshow test results in the results section, not only in methods.

Discussion (pp. 24–26, lines 321–395)

Comparisons with other studies are useful but lack depth. For example, explain why autonomy and choice remain low despite policy efforts (lines 355–358).

Avoid speculative statements without evidence (e.g., “medical error” as a reason for poor responsiveness, lines 372–374).

Strengthen explanation for non-significant socio-demographic factors—could sample homogeneity be a reason?

Provide concrete policy implications (e.g., how training and infrastructure improvements can address low autonomy and choice scores). You could have a section for this.

Expand on strengths and Limitations – The discussion should include methodological limitations, such as potential social desirability bias due to interviewer-administered surveys and facility-based sampling that may exclude home deliveries.

Conclusion (pp. 26–27, lines 396–410)

Avoid vague statements like “relatively higher” without benchmarks.

Include recommendation for integrating responsiveness indicators into routine maternal health quality assessments.

Strengthen Conclusion – While the conclusion summarizes key findings, it could better distinguish between empirical conclusions and recommendations, avoiding generalizations beyond collected data.

Language: Several grammatical issues and awkward phrasing (e.g., line 55, “crucial in impact health system responsiveness”). A thorough language edit is recommended.

**Do you want your identity to be public for this peer review?** For information about this choice, including consent withdrawal, please see our Privacy Policy

Reviewer #1: No

Reviewer #2: No

---

## [Author Response · Author response to Decision Letter 1]

10 Dec 2025

We would like to express our sincere appreciation to the reviewers for their valuable comments and constructive suggestions. We have carefully addressed all concerns raised during the review process and made the necessary revisions to strengthen the manuscript. A detailed response to each comment has also been provided as required.

Thank you once again for your time, guidance, and support throughout the review process.

---

## [Editor Report · Decision Letter 1]

25 Dec 2025

Health system responsiveness and its associated factors for Delivery Care in Public Health Facilities of West Arsi Zone, Oromia, Ethiopia

PONE-D-25-48434R1

Dear Dr. Gejo,

We’re pleased to inform you that your manuscript has been judged scientifically suitable for publication and will be formally accepted for publication once it meets all outstanding technical requirements.

Kind regards,

Engidaw Fentahun Enyew

Academic Editor

PLOS One
---

## [Editor Report · Acceptance letter]

PONE-D-25-48434R1

PLOS One

Dear Dr. Gejo,

I'm pleased to inform you that your manuscript has been deemed suitable for publication in PLOS One. Congratulations! Your manuscript is now being handed over to our production team.

Kind regards,

on behalf of

Mr Engidaw Fentahun Enyew

Academic Editor

PLOS One